# The Anti-Inflammatory, Anti-Oxidative, and Anti-Apoptotic Benefits of Stem Cells in Acute Ischemic Kidney Injury

**DOI:** 10.3390/ijms20143529

**Published:** 2019-07-19

**Authors:** Kuo-Hua Lee, Wei-Cheng Tseng, Chih-Yu Yang, Der-Cherng Tarng

**Affiliations:** 1Division of Nephrology, Department of Medicine, Taipei Veterans General Hospital, Taipei 11217, Taiwan; 2Institute of Clinical Medicine, National Yang-Ming University, Taipei 11217, Taiwan; 3Center for Intelligent Drug Systems and Smart Bio-devices (IDS2B), Hsinchu 30010, Taiwan; 4Department and Institute of Physiology, National Yang-Ming University, Taipei 11217, Taiwan

**Keywords:** ischemia-reperfusion, acute kidney injury, stem cell, conditioned medium, inflammation

## Abstract

Ischemia-reperfusion injury (IRI) plays a significant role in the pathogenesis of acute kidney injury (AKI). The complicated interaction between injured tubular cells, activated endothelial cells, and the immune system leads to oxidative stress and systemic inflammation, thereby exacerbating the apoptosis of renal tubular cells and impeding the process of tissue repair. Stem cell therapy is an innovative approach to ameliorate IRI due to its antioxidative, immunomodulatory, and anti-apoptotic properties. Therefore, it is crucial to understand the biological effects and mechanisms of action of stem cell therapy in the context of acute ischemic AKI to improve its therapeutic benefits. The recent finding that treatment with conditioned medium (CM) derived from stem cells is likely an effective alternative to conventional stem cell transplantation increases the potential for future therapeutic uses of stem cell therapy. In this review, we discuss the recent findings regarding stem cell-mediated cytoprotection, with a focus on the anti-inflammatory effects via suppression of oxidative stress and uncompromised immune responses following AKI. Stem cell-derived CM represents a favorable approach to stem cell-based therapy and may serve as a potential therapeutic strategy against acute ischemic AKI.

## 1. Introduction

Acute kidney injury (AKI) involves a complex interaction between the kidney parenchyma and immune system that leads to inflammation at the site of the injured tissue and impaired renal function [1]. Renal ischemia is a significant cause of AKI and is characterized by reduced tissue perfusion, which leads to acute tubular injury. Re-establishing the blood supply after prolonged ischemia activates vascular endothelial cells and enhances the generation of reactive oxygen species (ROS). This triggers a myriad of inflammatory consequences and induces apoptosis of tubular epithelial cells (TECs) [2]. This phenomenon is referred to as ischemia-reperfusion induced AKI (IR-AKI) and is characterized by elevated oxidative stress and activation of the immune system in response to ischemic tissue injury [3,4].

In the early stages of renal ischemia, the circulating neutrophils and monocytes rapidly infiltrate the ischemic kidney and release lysosomal enzymes, leading to tubular epithelial cell injury [2]. Subsequent crosstalk between the injured TECs, activated endothelial cells, and tissue macrophages induces oxidative stress and complement activation, aggravating cell damaging processes, such as mitochondrial dysfunction and lipid peroxidation [5].

Furthermore, the extensive release of pro-inflammatory cytokines, such as tumor necrosis factor (TNF)-α, interleukin (IL)-6, and monocyte chemoattractant protein 1 (MCP-1), attract an immune response involving monocytes, dendritic cells (DC), natural killer (NK) cells, and lymphocytes [6]. Subsequent ROS-mediated mitogen-activated protein kinase (MAPK) and nuclear factor (NF)-κB cascades amplify the inflammatory response. The signaling transduction pathways induce extensive TEC apoptosis and upregulate several important inflammatory mediators, such as IL-6, TNF-α, IL-1β, interferon (IFN)-γ, and IL-17 [2,4,7]. These immune reactions establish a continuous positive feedback loop, or a vicious circle, resulting in constant stimulation. As a consequence, IR-AKI is not merely a localized kidney insult, but also a trigger for a cascade of systemic inflammation (Figure 1).

In contrast to previous therapeutic approaches for AKI, which were mainly supportive, innovative treatments targeting AKI-induced inflammation, including stem cell therapy, have elicited a great deal of interest. In experimental models, administration of stem cells has proven effective in the treatment of AKI. One of the critical mechanisms of stem cell therapy is the anti-inflammatory effect caused by interaction with immune cells in the inflamed microenvironment [8,9,10]. Furthermore, stem cells may minimize the magnitude of tissue damage by secreting soluble cytoprotective factors in a paracrine manner [11,12].

Given that oxidative stress and inflammation have been implicated in the pathogenesis of IR-AKI, it is imperative to discuss these anti-inflammatory and immunoregulatory properties of stem cells. In this review, we focus on the therapeutic potential of stem cells in IR-AKI and illustrate the underlying antioxidant and anti-inflammatory mechanisms of this therapy. Because stem cells release soluble factors and microvesicles in a paracrine manner [13], we also discuss the effectiveness of stem cell-derived conditioned medium (CM) as an alternative to stem cell transplantation in the treatment of IR-AKI.

## 2. Immune Responses and Inflammation in IR-AKI

In IR-AKI, damaged cells are thought to be the critical trigger of inflammation. During the reperfusion phase, TECs are vulnerable to oxidative stress, and apoptotic and necrotic TECs release damage-associated molecular patterns (DAMPs) into the extracellular space. Endogenous DAMP molecules include DNA, RNA, and several intracellular proteins such as S100, heat-shock proteins, and high-mobility group box 1 (HMGB1) [14]. The so-called “danger signals” stimulate pattern recognition receptors (PRRs) expressed on renal parenchyma and immune cells, like epithelial and endothelial cells, DCs, lymphocytes, and macrophages. This recognition process initiates the host’s defense mechanisms and further produces various cytokines that attract neutrophils and macrophages [15]. Signaling pathways activated by DAMP ligation of PRRs also result in activation of NF-κB, which further promotes the expression of pro-inflammatory cytokines and perpetuates the inflammatory response in IR-AKI.

The balance between pro-inflammatory (e.g., TNF-α, IFN-γ, IL-6, IL-1β, IL-17, C3, C5a, and C5b) and anti-inflammatory (e.g., IL-4, TGF-β, IL-10, and heme oxygenase-1 (HO-1)) mediators secreted by the participating cell populations determines the status of injury and repair [16]. HO-1 is an endogenous stress-inducible enzyme, which modulates leukocyte adhesion and migration, immune cell maturation, and production of inflammatory cytokines following ischemia. Up-regulation of HO-1 represents an anti-inflammatory and anti-oxidative defense capacity against IRI [17]. Under ideal conditions, a regulated balance between inflammatory and anti-inflammatory mediators ensures healthy tissue regeneration and reversal of homeostatic conditions. However, AKI often results in impeded tissue repair attributed to sustained inflammation and secretion of profibrotic cytokines (e.g., IL-13 and TGF-β1), which triggers myofibroblast activation and progressive kidney fibrosis [18].

### Immunomodulatory Effects of Stem Cells

Considered an innovative anti-inflammatory treatment, the immunomodulatory effects of mesenchymal stem cells (MSCs) have been extensively investigated among other types of stem cells [19,20,21]. Firstly, MSCs are hypoimmunogenic as a result of reduced major histocompatibility complex (MHC) class I expression and a complete lack of expression of MHC class II and costimulatory molecules CD80 and CD86 [22]. This indicates that they can likely evade innate immunity processes, such as NK cell-mediated cytotoxicity, and lack the antigen presentation pathway essential for activation of the adaptive immune system [23]. To elicit an immunological balance, MSCs act as an immunomodulator by reducing the functional capacities and proliferation of all types of immune cells. MSCs have been proven to suppress lymphocyte activation and regulate their survival in a quiescent state [24]. Inhibition of T cell proliferation occurs through expression of inducible nitric oxide synthase (iNOS) and indoleamine 2,3-dioxygenase (IDO) in rodent and human MSCs, respectively [25]. More specifically, MSCs suppress CD4^+^ T helper (Th) cells from differentiating into their Th1 and Th17 subsets, which are the causal agents in the pathogenesis of autoimmunity [26]. On the other hand, MSCs enhance the proliferation of regulatory T cells (Tregs) and strengthen their immune modulating capacities [27,28]. MSCs also inhibit the differentiation, maturation, and activation of DCs by downregulating the surface expression of CD80, CD86, and MHC class II molecules, retaining the DCs in a tolerogenic phenotype. In this state, they express various factors, such as IDO and prostaglandin E2 (PGE2), which lower DC immunogenicity, reduce T cell proliferation, and induce Treg differentiation [29,30]. Simultaneously, MSCs induce macrophages to secrete immunosuppressive cytokines, like IL-4, IL-10 and transforming growth factor-β (TGF-β). It has also been shown that MSCs suppress NK cell proliferation and protect against perforin/granzyme-mediated cytotoxicity [31]. Furthermore, MSCs have inhibitory effects on B-cell proliferation, differentiation, and antibody production [32]. Given that HO-1 has significant anti-inflammatory therapeutic potential, recent research has pointed out HO-1-modified MSCs have an enhanced ability to attenuate inflammatory responses in ischemic heart disease [33], acute ischemic liver failure [34], lipopolysaccharide (LPS)-induced microvascular injury [35], and cisplatin-induced AKI [36]. Taken together, the administration of MSCs prevents immune cell activation and modulates kidney inflammation progression by managing cytokine secretion to promote anti-inflammatory processes.

It is noteworthy that the environment surrounding the MSCs is of critical importance to regulate the immunomodulatory effects. Liu et al. reported that MSCs derived from inflamed periodontal ligaments exhibit an impaired immunosuppressive capacity, with less inhibition of T cell proliferation, less induction of regulatory T cell, and less IL-10 production. The inflamed microenvironment also diminishes the immunomodulatory benefits of MSCs by reducing Th17 differentiation and IL-17 production [37]. Furthermore, Waterman et al. disclosed that MSCs could undergo functional polarization by differential Toll-like receptor (TLR) downstream signaling. Activation through TLR4 induced the pro-inflammatory MSC1, mostly producing pro-inflammatory mediators (IL-6, IL-8, and IFN-γ), can induce T cell activation. On the other hand, the TLR3-primed MSC2 mainly express anti-inflammatory factors such as IDO, PGE2, and HO-1, leading to T cell inhibition [38,39]. Moreover, to explain the diverse response of MSCs to TLR activation, Levin et al. suggested the level of co-cultured LPS-binding protein as a predictive factor in determining the secretomes of MSCs in response to TLR activation [40].

Similar to MSCs, Schnabel et al. found that induced pluripotent stem cells (iPSCs) possess immunomodulatory capacities evidenced by reducing responder T-cell proliferation in modified mixed leukocyte reactions in vitro [41]. Their findings echo our previous study [42], in which iPSCs without c-Myc were introduced into IR-AKI rat kidneys. This approach was not only safe, but also resulted in a substantial decrease in the levels of ROS and inflammatory cytokines. Furthermore, iPSCs have been shown to have strong immunomodulation effects through suppression of lymphocyte proliferation, NK cell-directed cytotoxicity, and DC differentiation and function [43,44,45]. This information is critical in considering the use of iPSCs in place of MSCs for both regenerative medicine and transplant medicine.

## 3. Oxidative Stress in IR-AKI

After the occurrence of acute ischemia, restoration of renal perfusion rapidly activates vascular endothelial cells, which trigger the production of pro-inflammatory cytokines and ROS, including superoxide (•O2^−^), hydrogen peroxide (HOOH), and hydroxyl radical (•OH) [46]. Following IR-AKI, defective antioxidant processes cause depletion of endogenous antioxidants and reduced activity of redox-regulated enzymes, exacerbating the accumulation of intracellular ROS. This increased ROS production associated with reduced antioxidant capacity leads to a state of oxidative stress, which ultimately results in mitochondrial damage, depletion of ATP, increased lipid peroxidation, and activation of cell death pathways. Another harmful effect of ROS is oxidative modification of cell membrane proteins; this impairs ion and nutrient transport, energy metabolism, and organelle function essential for cellular homeostasis [47]. Furthermore, ROS-mediated activation of NF-κB continues to exacerbate systemic inflammation, triggering TEC apoptosis and kidney fibrosis, which have a detrimental impact on renal function [48].

### Antioxidant Effects of Stem Cells

As a promising regenerative approach, stem cell therapy has been demonstrated to ameliorate various inflammatory diseases via its antioxidative activity [49,50,51,52]. MSCs can be isolated from bone marrow, umbilical cord blood, adipose tissue, placenta, periosteum, trabecular bone, synovium, skeletal muscle, and deciduous teeth [53], and their administration has been widely reported to upregulate the expression of the antioxidative enzyme HO-1 [35,54,55,56,57]. Increased HO-1 enzymatic activity is not only essential for MSC maturation, but is also cytoprotective against oxidative stress [58,59]. The antioxidant effects of HO-1 arise from its ability to increase reduced glutathione levels and degrade heme, as well as its ability to increase biliverdin and bilirubin, which have potent antioxidant properties [60,61]. In IR-AKI models, the increased production of HO-1 after MSC administration correlated with decreased levels of 8-hydroxy-2-deoxyguanosine (8-OHdG) and ROS [62]. The pro-angiogenic effects of MSCs lacking HO-1 expression are impaired; this triggers post-ischemic neovascularization and tissue repair, demonstrated by decreased secretion of several crucial pro-angiogenic growth factors, such as stromal cell-derived factor-1, vascular endothelial growth factor-A (VEGF-A), and hepatocyte growth factor (HGF) [36]. Similarly, CM derived from HO-1 knockout MSCs lacked therapeutic effects and failed to restore the functional and morphological changes in AKI [63]. A recent study also showed that modification with HO-1 significantly attenuated cell-cycle arrest, activated the PI3K/Akt and MEK/ERK pathways, and enhanced the survival of MSCs, all of which improved the therapeutic effects of MSCs against IR- AKI [64].

In addition to its impact on HO-1, IR-AKI also reduces the activity of antioxidant enzymes that scavenge ROS, including superoxide dismutase (SOD), catalase (CAT), glutathione-S-transferase (GST), and glutathione peroxidase (GPX), in post-ischemic kidney tissue [65]. MSC therapy increases the antioxidant capacity of post-ischemic kidney tissue by enhancing the activity of these ROS-scavenging enzymes, thereby reducing the levels of tissue malondialdehyde (MDA) [66,67,68]. Zhang et al. applied MSC-derived extracellular vesicles (MSC-EV) into an IR-AKI model and found that MSC-EV treatment reduced oxidative stress, and subsequently attenuated IR-AKI. This antioxidant effect is likely a result of activation of the NF-E2-related factor 2 (Nrf2)/antioxidant responsive element (ARE) pathway [69], but may also be due to decreased expression of NADPH oxidase 2 (NOX2) and ROS in injured kidney tissues [57,70].

In addition to MSCs, induced pluripotent stem cells (iPSC) are also equipped with antioxidative properties. In rats with IR- AKI, our previous study showed that the administration of iPSCs into kidneys via an intrarenal arterial route not only ameliorated the severity of tubular damage and kidney failure by reducing the expression of oxidative markers, pro-inflammatory cytokines, and apoptotic factors, but also improved the survival of IR-AKI rats [42]. We further showed that treatment of iPSC-CM in rats with IR-AKI significantly diminished oxidative stress and protected tubular cells against apoptosis [71], supporting the innovation occurring in this field of research.

## 4. Apoptosis of TEC in IR-AKI

Apoptosis is known to be a relevant mechanism of tubular cell death in IR-AKI. Kidney biopsies from IR-AKI animal models and humans have consistently shown apoptotic changes in TECs. There are several mechanisms of the pathogenesis of apoptosis of TECs. During ischemia, the pro-apoptotic protein Bax is upregulated in TECs, which results in a reduction of the anti-apoptotic protein Bcl-2, thus, promoting the initiation of apoptosis [72]. Another important stress kinase activated in the setting of ischemia is glycogen synthase kinase 3-β (GSK3β), which has been linked to mitochondrial dysfunction after exposure to oxidative stress [73]. During ischemia and ATP depletion, GSK3β upregulates Bax to activate caspase cascades, thus, promoting TEC apoptosis. Active GSK3β also positively regulates NF-κB leading to the inhibition of TNF-mediated apoptosis [74].

Other mechanisms of the activation of apoptotic pathways during IR-AKI have been proposed. The extrinsic apoptotic pathway is triggered by the binding of TNF-α and Fas ligands to death receptors, including Fas, tumor necrosis factor receptor 1 (TNFR1), and TNF-related apoptosis-inducing ligand receptors (TRAIL-Rs), expressed on TECs. Binding to these ligands results in receptor aggregation and recruitment of adaptor proteins, which, in turn, initiates a proteolytic cascade by activating initiator caspase-8 and caspase-10 [75]. The intrinsic apoptotic pathway is characterized by permeabilization of the mitochondrial outer membrane, resulting in the release of cytochrome c into the cytoplasm [76]. Cytochrome c then forms a multiprotein complex known as the “apoptosome” and initiates activation of the caspase cascade through caspase-9 [77]. Due to the significant consequences of TEC apoptosis and the complexity of its pathogenesis, reduction of tubular apoptosis is an essential requirement for the successful treatment of IR-AKI by stem cell therapy.

### Anti-Apoptotic Effects of Stem Cells

To date, the anti-apoptotic property of stem cells seems to be the most widely recognized beneficial effect of MSCs [49,78,79,80]. In experimental models of AKI, administration of MSCs displayed a renoprotective effect by preventing tissue apoptosis, which accelerated the repair of injured tissue. MSC-treated AKI mice showed increased expression of the anti-apoptotic gene BCL2 and downregulation of the pro-apoptotic gene BAX [81]. Regarding the mechanisms of these treatment processes, MSCs modulate tubular apoptosis and regeneration through production of soluble paracrine factors and trophic growth factors, including VEGF, HGF, insulin-like growth factor 1 (IGF-1), stanniocalcin-1, TGF-β, and fibroblast growth factor 2 [82,83]. Cumulating evidence indicates that MSCs release extracellular vesicles (EVs) that deliver genes, microRNAs, exosomes, and proteins to recipient cells, thus, acting as mediators of MSC paracrine action and conferring resistance to apoptosis [84,85]. These EVs are also thought to communicate intercellularly and influence the function of progenitor cells to stimulate angiogenesis and other reparative processes and, consequently, accelerate tissue repair [11,86].

iPSCs present a promising new therapeutic approach for AKI [87], and several studies have illustrated their anti-apoptotic effects against IR-AKI. Subcapsular transplantation of human-iPSCs in rodent kidneys attenuated TEC apoptosis and ameliorated histological alterations resulting in renal function improvements following IR-AKI [88]. Li et al. also demonstrated the therapeutic effect of iPSC-derived renal progenitor cells (RPC) in IR-AKI; they observed the reduction of tubular apoptosis and renal function recovery in a rat model of IR-AKI. Simultaneously, increased expression of anti-inflammatory mediators and growth factors involved in kidney repair were observed after transplantation of iPSC-derived RPCs and MSCs in injured kidneys [88,89,90]. Shen et al. also showed that iPSC-derived endothelial progenitor cells ameliorated apoptosis of TECs and cardiomyocytes while treating IR-AKI in mice [91]. Regarding the mechanisms of the anti-apoptotic properties of stem cells, our previous study suggested that iPSC-derived CM provided a protective effect against IR-AKI by reducing ROS generation, suppressing p38-MAPK activation, and inhibiting TNF-induced cell death and its downstream effect of NF-κB-induced systemic inflammation [71]. Therefore, we have demonstrated that iPSCs exerted renoprotective effects via the secretion of paracrine factors and suggest that iPSC-CM is a potential resource for stem cell-based therapy against IR-AKI.

The therapeutic potential of spermatogonial stem cells (SSC) in AKI has been explored in the preclinical setting. Unlike the production of soluble cytokines and growth factors by MSCs and iPSCs, the mechanism governing SSCs to accelerate tissue regeneration is through direct differentiation into renal parenchymal cells. To prove this phenomenon, Wu et al. injected mouse SSCs into adult female mice kidneys. Three months after SSC administration, the histological analysis revealed the transplanted SSCs migrated to the basement membrane and trans-differentiated into mature renal TECs. The most convincing evidence for self-renewal and multipotency of SSCs came from the presence of the Y chromosome in the nucleolus of TECs and glomerular podocytes isolated from the SSC-transplanted kidneys in female mice [92].

Under specific conditions, SSCs can spontaneously transform into germline cell-derived pluripotent stem cells (GPSCs), which can be readily frozen and thawed without loss of cell viability. Using a novel renal epithelial differentiation protocol, Chiara et al. generated GPSC-derived tubular-like cells (GTCs) resembling renal TEC phenotypes and biological functions. After administration of GTCs intravenously in IR-AKI mice, these cells were able to home in on sites of inflammation and showed long-term engraftment in the injured kidney. Histological analysis disclosed less extent of cortical damage, inflammatory infiltrate, and interstitial fibrosis in the GTC-treated kidney. The GTCs also elicit cytoprotective functions in reducing renal oxidative stress, tubular apoptosis, and upregulation tubular expression of HO-1. Accordingly, GPSCs could be considered as a potential stem cell therapy against IR-AKI and subsequent chronic kidney damage [93].

## 5. Stem Cells in the Context of Clinical Use

Clinical applications of stem cell therapy are widely under investigation, as they possess anti-inflammatory, anti-fibrotic, and anti-apoptotic properties. However, clinical trials evaluating the therapeutic potentials of stem cells in AKI are still in the evolving stage, and their promise in preclinical models is yet to be translated. Dating back to 2008, the first phase 1 clinical trial (NTC00733876) evaluated the safety and efficacy of MSCs in AKI initiated with open-heart surgery. This study enrolled 16 open-heart surgical patients, and bone marrow-derived MSCs were administered into the suprarenal aorta through a femoral catheter after completion of surgery. The inclusion criteria were patients at high risk for postoperative AKI, such as old age, underlying diabetes mellitus, congestive heart failure, chronic obstructive lung disease, and pre-existing CKD stage 1-4. The exclusion criteria were active infection, evolving myocardial infarction, cardiogenic shock, history of malignancy, or advanced CKD stage 5/5D. The primary outcome was the absence of MSC-specific adverse events. During the six-month follow-up period, there were no specific or serious adverse effects observed, and this study concluded that infusions of MSCs might provide a novel and safe approach for inducing renal protection [94]. Based on this positive result, a subsequent multicenter randomized controlled trial in 2017 (NCT01602328) was conducted to determine the efficacy of allogeneic human MSCs in accelerating kidney recovery from established AKI. This phase 2 study enrolled patients who developed AKI within 48 h after cardiac surgery, and they randomized a total 156 participants to receive allogeneic MSCs (AC607, in a single dose of 2 × 10^6^ cells/kg) or placebo administration through an intra-aortic route. The primary outcome was the time to recovery of kidney function. At the end of follow-up, although treatment with MSCs was found to be safe and tolerated well, this study concluded that administration of MSCs did not decrease the time to renal function recovery or provision for dialysis. Besides, the 30-day all-cause mortality was comparable between MSCs group and placebo group, and the rates of other major adverse kidney events were similar [95]. From these two early-phase clinical trials, the role of administering allogeneic MSCs for postcardiac surgery AKI is initially recognized. Although MSCs may be of no value as a therapy to recover renal function in established AKI, the preliminary analysis showed that MSC administration is safe at all tested doses. Unfortunately, there are no other ongoing registered clinical trials for the treatment of postcardiac surgery AKI, thus leaving unexplored the possibility of a potential beneficial effect of MSC therapy at doses higher than those reported so far.

Other clinical trials regarding AKI situation include administration of MSC to kidney transplant recipients. In a single-site, prospective, open-label, randomized study in China (NCT00658073), a total of 159 adult subjects underwent kidney transplants with allografts from living donors were divided into three groups: The standard dose (*n* = 53) and lower dose (80% of standard, *n* = 52) calcineurin inhibitors (CNI), in combination with a double intravenous infusion of autologous bone marrow-derived MSCs (1–2 × 10^6^/kg) at kidney reperfusion and 2 weeks later. Patients (*n* = 51) in the control group were given the anti-IL-2 receptor antibody basiliximab induction therapy, plus standard dose CNI. The main outcome included the one-year incidence of acute rejection, adverse events, patient and graft survival. Compared to the basiliximab group, this study demonstrated that the use of autologous MSC resulted in a lower incidence of acute rejection, lower risk of opportunistic infection, and better graft function at one year [96]. Another trial also suggested MSCs enable 50% reduction of CNI maintenance immunosuppression in living donor kidney transplant recipients [97]. Therefore, MSC-based therapy has proven to reduce induction and maintenance of immunosuppressive drugs without compromising patient safety and graft outcome. This may be due to the immunomodulatory activity of MSCs, but these studies, unfortunately, did not address the underlying mechanism.

A clinical trial using stem cells in treating AKI receiving continuous renal replacement therapy (CRRT) is ongoing (NCT03015623) [98]. AKI participants were treated with extracorporeal therapy with hemofiltration device containing millions of allogeneic MSCs (SBI-101) up to 24 h, designed to regulate inflammation and promote repair of injured tissue. Instead of intravenous infusion of allogeneic MSCs that are diluted rapidly throughout the body, SBI-101 allows delivery of a stable dose of cells by exposing the blood ultrafiltrate to MSCs that are immobilized on the extraluminal side of membranes within the hollow fiber dialyzer. This provides AKI patients with both standard-of-care hemofiltration as well as MSC-mediated blood conditioning in a single session. The conditioned ultrafiltrate is then delivered back to the subject, which allows for continuous exposure of the MSCs to patient blood during the CRRT treatment. In this trial, the recruitment is currently active, and subjects will be randomized into three different doses: Low dose SBI-101 containing 250 million MSCs, high dose SBI-101 containing 750 million MSCs, or sham control to characterize the pharmacokinetics and pharmacodynamics of SBI-101. In this first-in-human clinical trial, the primary outcome is its safety and tolerability. Measures of SBI-101 efficacy could be reduced patient time on dialysis or reduced patient time in the ICU.

There are still some barriers in the utilization of stem cells in clinical settings for AKI. Although MSC therapy has multiple benefits with no detrimental side effects, so far it still lacks both long-term follow-up data and the consensus in therapeutic protocols. Furthermore, the collection of MSCs from bone marrow is relatively invasive and the source is not available in a large volume. Similarly, SSC-based therapies in AKI have some limitations. Although SSCs are recognized to differentiate into renal lineages, their promise in preclinical AKI models is not yet translated in humans. Furthermore, even though SSCs can be administered in both genders, they can only be harvested from the testis and require a somewhat invasive procedure on male donors. In regard to iPSCs, c-Myc, one of the reprogramming factors to induce pluripotency, is a well-known oncogene leading to tumorigenesis. Therefore, the adverse effect of teratoma or tumor formation derived from iPSC treatment warrants significant concern. Our previous study demonstrated that rats treated with iPSCs without c-Myc effectively blocked the teratoma formation [42]. Alternatively, therapy utilizing iPSC-CM showed the promising anti-inflammatory benefits for IR-AKI and eliminated the concern of tumorigenesis as well [71]. Until now, there are few clinical trials of iPSC or stem cell-derived CM containing soluble factors and EVs in the treatment of AKI, and the future outcomes are highly expected [99].

## 6. Conclusions

In summary, animal experiments have provided compelling evidence to support a renoprotective role for stem cells in rescuing IR-AKI. Multiple mechanisms have been proposed to explain the beneficial effects of stem cells and their derived CM, including antioxidant, immunomodulatory, and anti-apoptotic effects (Figure 2). Another essential component of the beneficial effects of stem cells is their production of soluble paracrine factors and trophic growth factors. Moreover, recent investigations have found that stem cell-derived EVs may carry pro-regenerative micro-RNA molecules that stabilize vascular and tubular function, which has therapeutic potential for rescuing IR-AKI. Although the majority of studies in the field of IR-AKI show remarkable benefits of stem cell therapy, they are mostly confined to experimental animal models. More translational studies are needed to provide a more comprehensive understanding of stem cell-based therapies and to ensure their safety for future clinical applications.

## Figures and Tables

**Figure 1 ijms-20-03529-f001:**
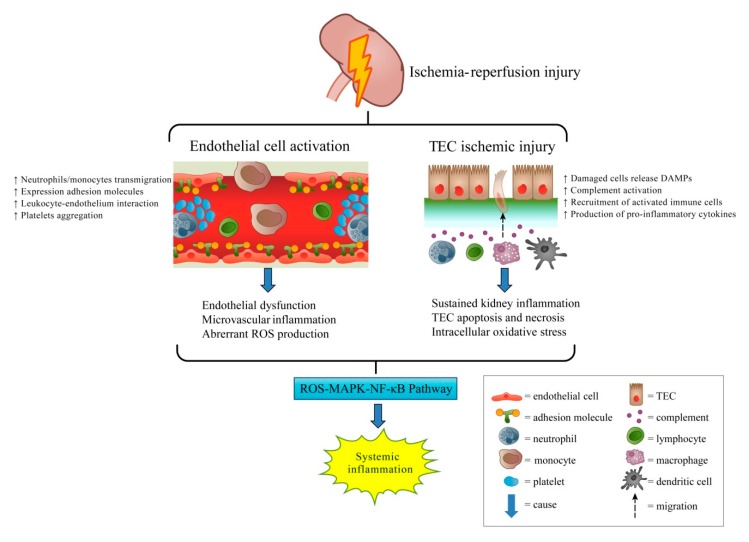
Pathogenesis of ischemia-reperfusion induced acute kidney injury. Ischemia-reperfusion-induced acute kidney injury involves endothelial cell activation (1). The increased leukocyte adhesion molecules on the activated endothelial cells induce leukocyte transmigration and platelet aggregation, which both cause microvascular inflammation. In tubular epithelial cell injury (2), the injured tubular cells release danger signals, which activate immune cells involved in local and systemic inflammation. The substantial amount of reactive oxygen species generated by this process activates the MAPK-NF-κB pathway and induces systemic inflammation. Abbreviations: DAMPs, damage-associated molecular patterns; TECs, tubular epithelial cells; ROS, reactive oxygen species; MAPK, mitogen-activated protein kinase; NF-κB, nuclear factor-κB.

**Figure 2 ijms-20-03529-f002:**
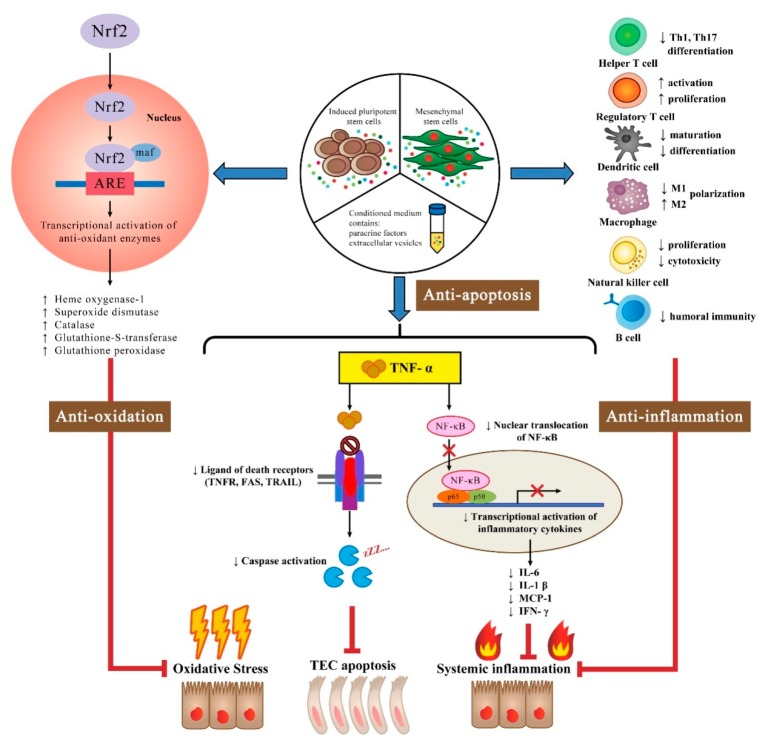
Illustration of proposed mechanisms of stem cell therapy in rescuing ischemia-reperfusion induced acute kidney injury. The therapeutic effects of mesenchymal stem cells, induced pluripotent stem cells, and their conditioned medium containing soluble factors and extracellular vesicles include: (1) Anti-oxidation, which may act through activation of the Nrf2/ARE pathway and, subsequently, upregulation of antioxidative enzymes against oxidative stress; (2) anti-inflammation, via immunosuppressive effects on immune cells and inhibition of NF-κB transcriptional activity; and (3) anti-apoptosis, possibly through decreased tumor necrosis factor-induced intrinsic apoptosis signaling. Abbreviations: Nrf2, NF-E2-related factor 2; ARE, antioxidant responsive element; TNF-α, tumor necrosis factor-α; TNFR, tumor necrosis factor receptor; TRAIL, TNF-related apoptosis-inducing ligand; TEC, tubular epithelial cell; NF-κB, nuclear factor-κB; MCP-1, monocyte chemoattractant protein 1; IFN-γ, Interferon-γ.

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
