# Peer review of "The Anti-Inflammatory, Anti-Oxidative, and Anti-Apoptotic Benefits of Stem Cells in Acute Ischemic Kidney Injury"

_ijms, 2019, doi:10.3390/ijms20143529_

Round 1

Reviewer 1 Report

The review paper by Lee et al. provides a complete overview of the use of mesenchymal stem cells (or induced pluripotent stem cells) to potentially treat acute kidney injury due to ischemia. The paper provides an excellent summary of basic principles and highlights the need for clinical trials using stem cells in acute kidney injury.

Major points:
- I would first discuss “immune responses and inflammation in IR-AKI” (part 3) and then “oxidative stress in IR-AKI” (part 2); so switch parts 3 and 2.

- Is anything known on how exactly HO-1 is anti-inflammatory? Is it just because it is anti-oxidative?

- Can stem cells become proinflammatory due to the inflamed microenvironment? Are there any reports of this? 

Minor points:

- IR-AKI and IR-induced AKI are used interchangeably; only using the first one suffices.

- Line 166: I would remove “M2” and leave the cytokines that these macrophages produce; the field is slowly changing toward a different description of macrophages other than M1/M2.

- Line 235: please rephrase, this sentence is difficult to understand

- Line 245: remove “the” before active infection

- Line 250: randomized “controlled” trial, please correct

- Line 260: “compatible” should be “comparable”

- Line 268: remove “the” just before kidney transplant recipients

- Line 285: replace “underwent” with “ongoing”; add “were” just before treated

- Line 420: reference 31, please remove “original article” from the title of the citation

Author Response

Response to Reviewer 1 Comments

The review paper by Lee et al. provides a complete overview of the use of mesenchymal stem cells (or induced pluripotent stem cells) to potentially treat acute kidney injury due to ischemia. The paper provides an excellent summary of basic principles and highlights the need for clinical trials using stem cells in acute kidney injury.

Major Points:

Point 1: I would first discuss “immune responses and inflammation in IR-AKI” (part 3) and then “oxidative stress in IR-AKI” (part 2); so switch parts 3 and 2.

Response: Thank you for your suggestion. We have switched the 2 paragraphs (parts 2 and 3) in the text.

Point 2: Is anything known on how exactly HO-1 is anti-inflammatory? Is it just because it is anti-oxidative?

Response: We appreciate your great comment. You are correct that HO-1 not only reduces the oxidative stress but also equips anti-inflammatory therapeutic potential.

We had added the following descriptions in the paragraph " Immune Responses and Inflammation in IR-AKI".

Line 94~97:

“HO-1 is an endogenous stress-inducible enzyme, which modulates leukocyte adhesion and migration, immune cell maturation, and production of inflammatory cytokines following ischemia. Up-regulation of HO-1 represents an anti-inflammatory and anti-oxidative defense capacity against IRI.”

Line126~130:

“Given that HO-1 has significant anti-inflammatory therapeutic potential, recent research has pointed out HO-1-modified MSCs have an enhanced ability to attenuate inflammatory responses in ischemic heart disease, acute ischemic liver failure, lipopolysaccharide (LPS)-induced microvascular injury, and cisplatin-induced AKI”

Point 3: Can stem cells become pro-inflammatory due to the inflamed microenvironment? Are there any reports of this?

Response: We appreciate your great comment regarding the impact of inflammation on stem cell functions. We had added the following descriptions in the paragraph " Immune Responses and Inflammation in IR-AKI".

Line133~145:

It is noteworthy that the environment surrounding the MSCs is of critical importance to regulate the immunomodulatory effects. Liu et al. reported that MSCs derived from inflamed periodontal ligaments exhibit an impaired immunosuppressive capacity, with less inhibition of T cell proliferation, less induction of regulatory T cell, and less IL-10 production. The inflamed microenvironment also diminishes the immunomodulatory benefits of MSCs by reducing Th17 differentiation and IL17 production. Furthermore, Waterman et al. disclosed that MSCs could undergo functional polarization by differential Toll-like receptor (TLR) downstream signaling. Activation through TLR4 induced the pro-inflammatory MSC1, mostly producing pro-inflammatory mediators (IL-6, IL-8, and IFN-γ) and can induce T cell activation. On the other hand, the TLR3-primed MSC2 mainly express anti-inflammatory factors such as IDO, PGE2, and HO-1, leading to T cell inhibition. Moreover, to explain the diverse response of MSCs to TLR activation, Levin et al. suggested the level of co-cultured LPS-binding protein as a predictive factor in determining the secretomes of MSCs in response to TLR activation”

Minor points:

Point 1: IR-AKI and IR-induced AKI are used interchangeably; only using the first one suffices.

Response: Thank you for your suggestion. These wording changes has been corrected.

Point 2: I would remove “M2” and leave the cytokines that these macrophages produce; the field is slowly changing toward a different description of macrophages other than M1/M2.

Response: Thank you for your suggestion. We have removed the "M2" description of macrophages and leave the anti-inflammatory cytokines in the text.

Line122~123:

“Simultaneously, MSCs induce macrophages to secrete immunosuppressive cytokines, like IL-4, IL10 and transforming growth factorβ (TGF-β)”

Point 3 Line 235: please rephrase, this sentence is difficult to understand

Response: Thank you for your suggestion. This sentence has been changed as suggested to improve clarity

Line257~259:

“Therefore, we have demonstrated that iPSCs exerted renoprotective effects via the secretion of paracrine factors and suggested that iPSC-CM is a potential resource for stem cell-based therapy against IR-AKI.”

Point 4 Line 245: remove “the” before active infection

Response: This has been corrected.

Point 5 Line 250: randomized “controlled” trial, please correct

Response: This has been corrected.

Point 6 Line 260: “compatible” should be “comparable”

Response: This has been corrected.

Point 7 Line 268: remove “the” just before kidney transplant recipients

Response: This has been corrected.

Point 8 Line 285: replace “underwent” with “ongoing”; add “were” just before treated

Response: This has been corrected.

Point 9 Line 420: reference 31, please remove “original article” from the title of the citation

Response: This has been corrected.

Reviewer 2 Report

The authors did an intensive literature review on the selected topic. The text is well written and comprehensive to the readers. The reviewer have few minor suggestions that could improve the manuscript.

1)      The author highlighted the different beneficial effects of stem cells includes iPSCs and MSCs on AKI, but the text does not include SSC (spermatogonial stem cells), which is important as SSCs has several pre-clinical benefits for treatment of AKI.

2)      In addition, the text should include the limitations of use of each type of stem cell on treatment of AKI, since not all stem cells has similar properties.

3)      Error at line 50, ‘interferon (INF)-γ’ should be ‘interferon (IFN)-γ’

4)      Line 233, the content in this topic is very dense, could be shortened or simplified

5)      The manuscript title could be more informative, as the text also talks about anti-oxidation, anti-apoptotic benefits of stem cells.

The author used sufficient references.

Author Response

Response to Reviewer 2 Comments

The authors did an intensive literature review on the selected topic. The text is well written and comprehensive to the readers. The reviewer have few minor suggestions that could improve the manuscript.

Point 1: The author highlighted the different beneficial effects of stem cells includes iPSCs and MSCs on AKI, but the text does not include SSC (spermatogonial stem cells), which is important as SSCs has several pre-clinical benefits for treatment of AKI.

Response: We agree with your comments and have added the paragraph about the use of spermatogonial stem cells in AKI.

Line 260~268:

“ The therapeutic potential of spermatogonial stem cells (SSC) in AKI has been explored in the preclinical setting. Unlike the production of soluble cytokines and growth factors by MSCs and iPSCs, the mechanism governing SSCs to accelerate tissue regeneration is through direct differentiation into renal parenchymal cells. To prove this phenomenon, Wu et al. injected the mouse SSCs into adult female mice kidneys. Three months after SSC administration, the histological analysis revealed the transplanted SSCs migrated to the basement membrane and trans-differentiated into mature renal TECs. The most convincing evidence for self-renewal and multipotency of SSCs came from the presence of the Y chromosome in the nucleolus of TECs and glomerular podocytes isolated from the SSC-transplanted kidneys in female mice.”

Line 269~279:

“Under specific conditions, SSCs can spontaneously transform into germline cell-derived pluripotent stem cells (GPSCs), which can be readily frozen and thawed without loss of cell viability. Using a novel renal epithelial differentiation protocol, Chiara et al. generated GPSC-derived tubular-like cells (GTCs) resembling renal TEC phenotypes and biological functions. After administration GTCs intravenously in IR-AKI mice, these cells were able to home to sites of inflammation and showed long-term engraftment in the injured kidney. Histological analysis disclosed less extent of cortical damage, inflammatory infiltrate, and interstitial fibrosis in the GTC-treated kidney. The GTCs also elicit cytoprotective functions in reducing renal oxidative stress, tubular apoptosis, and upregulation tubular expression of HO-1. Accordingly, GPSCs could be considered as a potential stem cell therapy against IR-AKI and subsequent chronic kidney damage.”

Point 2: In addition, the text should include the limitations of use of each type of stem cell on treatment of AKI, since not all stem cells has similar properties.

Response: We agree with your comments and have added the paragraph about the limitations of each type of stem cells in the manuscript.

Line 342~357:

“There are still some barriers in the utilization of stem cells in clinical settings for AKI. Although MSC therapy has multiple benefits with no detrimental side effects, so far it still lacks both long-term follow-up data and the consensus in therapeutic protocols. Furthermore, the collection of MSCs from bone marrow is relatively invasive and the source is not available in a large volume. Similarly, SSC-based therapies in AKI have some limitations. Although SSCs are recognized to differentiate into renal lineages, their promising in preclinical AKI models is not yet translated in human. Furthermore, even though SSCs can be administered in both genders, they can only be harvested from the testis and require a somewhat invasive procedure on male donors. In regard to iPSCs, c-Myc, one of the reprogramming factors to induce pluripotency, is a well-known oncogene leading to tumorigenesis. Therefore, the adverse effect of teratoma or tumor formation derived from iPSC treatment warrants significant concern. Our previous study demonstrated that rats treated with iPSCs without c-Myc effectively blocked the teratoma formation [42]. Alternatively, therapy utilizing iPSC-CM showed the promising anti-inflammatory benefits for IR-AKI and eliminated the concern of tumorigenesis as well [71]. Till now, there are few clinical trials of iPSC or stem cell-derived CM containing soluble factors and EVs in the treatment of AKI, and the future outcomes are highly expected. ”

Point 3: Error at line 50, ‘interferon (INF)-γ’ should be ‘interferon (IFN)-γ’

Response: The error has been corrected.

Point 4: Line 235, the content in this topic is very dense, could be shortened or simplified

Response: As suggested by the reviewer, we have eliminated unnecessary words in this topic.

Point 5: The manuscript title could be more informative, as the text also talks about anti-oxidation, anti-apoptotic benefits of stem cells.

Response: As suggested by the reviewer, we have changed the title to “The Anti-inflammatory, Anti-oxidative and Anti-apoptotic Benefits of Stem Cells in Acute Ischemic Kidney Injury”

This manuscript is a resubmission of an earlier submission. The following is a list of the peer review reports and author responses from that submission.